# REAL OR NOT REAL, THAT IS THE QUESTION

**Yuanbo Xiangli**[1]*,   **Yubin Deng**[1]*,   **Bo Dai**[1]*,   **Chen Change Loy**[2],   **Dahua Lin**[1]
The Chinese University of Hong Kong          Nanyang Technological University
{xy019,dy015,bdai,dhlin}@ie.cuhk.edu.hk   ccloy@ntu.edu.sg

## ABSTRACT

While generative adversarial networks (GAN) have been widely adopted in various topics, in this paper we generalize the standard GAN to a new perspective by treating realness as a random variable that can be estimated from multiple angles. In this generalized framework, referred to as RealnessGAN[1], the discriminator outputs a distribution as the measure of realness. While RealnessGAN shares similar theoretical guarantees with the standard GAN, it provides more insights on adversarial learning. Compared to multiple baselines, RealnessGAN provides stronger guidance for the generator, achieving improvements on both synthetic and real-world datasets. Moreover, it enables the basic DCGAN (Radford et al., 2015) architecture to generate realistic images at 1024*1024 resolution when trained from scratch.

## 1   INTRODUCTION

The development of generative adversarial network (GAN) (Goodfellow et al., 2014; Radford et al., 2015; Arjovsky et al., 2017) is one of the most important topics in machine learning since its first appearance in (Goodfellow et al., 2014). It learns a discriminator along with the target generator in an adversarial manner, where the discriminator distinguishes generated samples from real ones. Due to its flexibility when dealing with high dimensional data, GAN has obtained remarkable progresses on realistic image generation (Brock et al., 2019).

In the standard formulation (Goodfellow et al., 2014), the realness of an input sample is estimated by the discriminator using a *single scalar*. However, for high dimensional data such as images, we naturally perceive them from more than one angles and deduce whether it is life-like based on multiple criteria. As shown in Fig.1, when a portrait is given, one might focus on its facial structure, skin tint, hair texture and even details like iris and teeth if allowed, each of which indicates a different aspect of realness. Based on this observation, the single scalar could be viewed as an abstract or a summarization of multiple measures, which together reflect the overall realness of an image. Such a concise measurement may convey insufficient information to guide the generator, potentially leading to well-known issues such as mode-collapse and gradient vanishing.

In this paper, we propose to generalize the standard framework (Goodfellow et al., 2014) by treating realness as a random variable, represented as a distribution rather than a single scalar. We refer to

---

*Equal contribution.
[1]Code will be available at https://github.com/kam1107/RealnessGAN

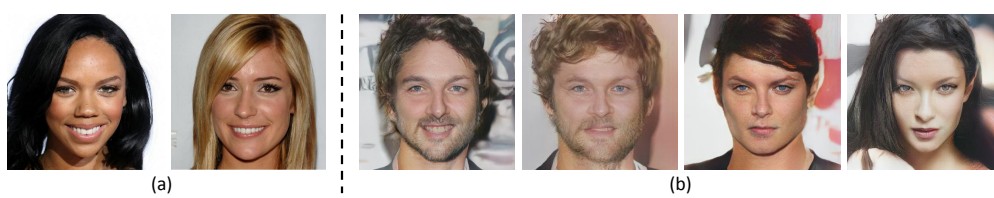

(a)                                                                 (b)

Figure 1: The perception of realness depends on various aspects. (a) Human-perceived flawless. (b) Potentially reduced realness due to: inharmonious facial structure/components, unnatural background, abnormal style combination and texture distortion.

such a generalization as RealnessGAN. The learning process of RealnessGAN abide by the standard setting, but in a distributional form. While the standard GAN can be viewed as a special case of RealnessGAN, RealnessGAN and the standard GAN share similar theoretical guarantees. *i.e.* RealnessGAN converges to a Nash-equilibrium where the generator and the discriminator reach their optimalities. Moreover, by expanding the scalar realness score into a distributional one, the discriminator $D$ naturally provides stronger guidance to the generator $G$ where $G$ needs to match not only the overall realness (as in the standard GAN), but the underlying realness distribution as well. Consequently, RealnessGAN facilitates $G$ to better approximate the data manifold while generating decent samples. As shown in the experiments, based on a rather simple DCGAN architecture, RealnessGAN could successfully learn from scratch to generate realistic images at 1024*1024 resolution.

## 2 REALNESSGAN

### 2.1 GENERATIVE ADVERSARIAL NETWORKS

Generative adversarial network jointly learns a generator $G$ and a discriminator $D$, where $G$ attempts to generate samples that are indistinguishable from the real ones, and $D$ classifies generated and real samples. In the original work of (Goodfellow et al., 2014), the learning process of $D$ and $G$ follows a minimax game with value function $V(G, D)$:

$$\min_{G} \max_{D} V(G, D) = \mathbb{E}_{\boldsymbol{x} \sim p_{\text{data}}}[\log D(\boldsymbol{x})] + \mathbb{E}_{\boldsymbol{z} \sim p_{\boldsymbol{z}}}[\log(1 - D(G(\boldsymbol{z})))], \quad (1)$$

$$= \mathbb{E}_{\boldsymbol{x} \sim p_{\text{data}}}[\log(D(\boldsymbol{x}) - 0)] + \mathbb{E}_{\boldsymbol{x} \sim p_g}[\log(1 - D(\boldsymbol{x}))], \quad (2)$$

where the approximated data distribution $p_g$ is defined by a prior $p_{\boldsymbol{z}}$ on input latent variables and $G$. As proved by Goodfellow et al. (2014), under such a learning objective, the optimal $D$ satisfies $D_G^*(\boldsymbol{x}) = \frac{p_{\text{data}}(\boldsymbol{x})}{p_{\text{data}}(\boldsymbol{x}) + p_g(\boldsymbol{x})}$ for a fixed $G$. Fixing $D$ at its optimal, the optimal $G$ satisfies $p_g = p_{\text{data}}$. The theoretical guarantees provide strong supports for GAN's success in many applications (Radford et al., 2015; Yu et al., 2017; Zhu et al., 2017; Dai et al., 2017), and inspired multiple variants (Arjovsky et al., 2017; Mao et al., 2017; Zhao et al., 2017; Berthelot et al., 2017) to improve the original design. Nevertheless, a *single scalar* is constantly adopted as the measure of realness, while the concept of realness is essentially a random variable covering multiple factors, *e.g.* texture and overall configuration in the case of images. In this work, we intend to follow this observation, encouraging the discriminator $D$ to learn a realness distribution.

### 2.2 A DISTRIBUTIONAL VIEW ON REALNESS

We start by substituting the scalar output of a discriminator $D$ with a distribution $p_{\text{realness}}$, so that for an input sample $\boldsymbol{x}$, $D(\boldsymbol{x}) = \{p_{\text{realness}}(\boldsymbol{x}, u); u \in \Omega\}$, where $\Omega$ is the set of outcomes of $p_{\text{realness}}$. Each outcome $u$ can be viewed as a potential realness measure, estimated via some criteria. While 0 and 1 in equation 2 are used as two virtual ground-truth scalars that respectively represent the realness of real and fake images, we also need two virtual ground-truth distributions to stand for the realness distributions of real and fake images. We refer to these two distributions as $\mathcal{A}_1$ (real) and $\mathcal{A}_0$ (fake), which are also defined on $\Omega$. As in the standard GAN where 0 and 1 can be replaced with other scalars such as $-1$ and 1, there are various choices for $\mathcal{A}_1$ and $\mathcal{A}_0$. Factors lead to a good pair of $\mathcal{A}_1$ and $\mathcal{A}_0$ will be discussed later. Accordingly, the difference between two scalars is replaced with the Kullback-Leibler (KL) divergence. The minimax game between a generator $G$ and a distributional discriminator $D$ thus becomes

$$\max_{G} \min_{D} V(G, D) = \mathbb{E}_{\boldsymbol{x} \sim p_{\text{data}}}[\mathcal{D}_{\text{KL}}(\mathcal{A}_1 \| D(\boldsymbol{x}))] + \mathbb{E}_{\boldsymbol{x} \sim p_g}[\mathcal{D}_{\text{KL}}(\mathcal{A}_0 \| D(\boldsymbol{x}))]. \quad (3)$$

An immediate observation is that if we let $p_{\text{realness}}$ be a discrete distribution with two outcomes $\{u_0, u_1\}$, and set $\mathcal{A}_0(u_0) = \mathcal{A}_1(u_1) = 1$ and $\mathcal{A}_0(u_1) = \mathcal{A}_1(u_0) = 0$, the updated objective in equation 3 can be explicitly converted to the original objective in equation 2, suggesting RealnessGAN is a generalized version of the original GAN.

Following this observation, we then extend the theoretical analysis in Goodfellow et al. (2014) to the case of RealnessGAN. Similar to Goodfellow et al. (2014), our analysis concerns the space of

probability density functions, where $D$ and $G$ are assumed to have infinite capacities. We start from finding the optimal realness discriminator $D$ for any given generator $G$.

**Theorem 1.** *When $G$ is fixed, for any outcome $u$ and input sample $x$, the optimal discriminator $D$ satisfies*

$$D_G^\star(\boldsymbol{x}, u) = \frac{\mathcal{A}_1(u)p_{data}(\boldsymbol{x}) + \mathcal{A}_0(u)p_g(\boldsymbol{x})}{p_{data}(\boldsymbol{x}) + p_g(\boldsymbol{x})}. \tag{4}$$

*Proof.* Given a fixed $G$, the objective of $D$ is:

$$\min_D V(G, D) = \mathbb{E}_{\boldsymbol{x} \sim p_{\text{data}}}[\mathcal{D}_{\text{KL}}(\mathcal{A}_1 \| D(\boldsymbol{x}))] + \mathbb{E}_{\boldsymbol{x} \sim p_g}[\mathcal{D}_{\text{KL}}(\mathcal{A}_0 \| D(\boldsymbol{x}))], \tag{5}$$

$$= \int_{\boldsymbol{x}} \left( p_{\text{data}}(\boldsymbol{x}) \int_u \mathcal{A}_1(u) \log \frac{\mathcal{A}_1(u)}{D(\boldsymbol{x}, u)} du + p_g(\boldsymbol{x}) \int_u \mathcal{A}_0(u) \log \frac{\mathcal{A}_0(u)}{D(\boldsymbol{x}, u)} du \right) dx, \tag{6}$$

$$= - \int_{\boldsymbol{x}} (p_{\text{data}}(\boldsymbol{x})h(\mathcal{A}_1) + p_g(\boldsymbol{x})h(\mathcal{A}_0)) \, dx$$
$$- \int_{\boldsymbol{x}} \int_u (p_{\text{data}}(\boldsymbol{x})\mathcal{A}_1(u) + p_g(\boldsymbol{x})\mathcal{A}_0(u)) \log D(\boldsymbol{x}, u) du dx, \tag{7}$$

where $h(\mathcal{A}_1)$ and $h(\mathcal{A}_0)$ are their entropies. Marking the first term in equation 7 as $C_1$ since it is irrelevant to $D$, the objective thus is equivalent to:

$$\min_D V(G, D) = - \int_{\boldsymbol{x}} (p_{\text{data}}(\boldsymbol{x}) + p_g(\boldsymbol{x})) \int_u \frac{p_{\text{data}}(\boldsymbol{x})\mathcal{A}_1(u) + p_g(\boldsymbol{x})\mathcal{A}_0(u)}{p_{\text{data}}(\boldsymbol{x}) + p_g(\boldsymbol{x})} \log D(\boldsymbol{x}, u) du dx + C_1, \tag{8}$$

where $p_{\boldsymbol{x}}(u) = \frac{p_{\text{data}}(\boldsymbol{x})\mathcal{A}_1(u) + p_g(\boldsymbol{x})\mathcal{A}_0(u)}{p_{\text{data}}(\boldsymbol{x}) + p_g(\boldsymbol{x})}$ is a distribution defined on $\Omega$. Let $C_2 = p_{\text{data}}(\boldsymbol{x}) + p_g(\boldsymbol{x})$, we then have

$$\min_D V(G, D) = C_1 + \int_{\boldsymbol{x}} C_2 \left( - \int_u p_{\boldsymbol{x}}(u) \log D(\boldsymbol{x}, u) du + h(p_{\boldsymbol{x}}) - h(p_{\boldsymbol{x}}) \right) dx, \tag{9}$$

$$= C_1 + \int_{\boldsymbol{x}} C_2 \mathcal{D}_{\text{KL}}(p_{\boldsymbol{x}} \| D(\boldsymbol{x})) dx + \int_{\boldsymbol{x}} C_2 h(p_{\boldsymbol{x}}) dx. \tag{10}$$

Observing equation 10, one can see that for any valid $\boldsymbol{x}$, when $\mathcal{D}_{\text{KL}}(p_{\boldsymbol{x}} \| D(\boldsymbol{x}))$ achieves its minimum, $D$ obtains its optimal $D^\star$, leading to $D^\star(\boldsymbol{x}) = p_{\boldsymbol{x}}$, which concludes the proof. $\square$

Next, we move on to the conditions for $G$ to reach its optimal when $D = D_G^\star$.

**Theorem 2.** *When $D = D_G^\star$, and there exists an outcome $u \in \Omega$ such that $\mathcal{A}_1(u) \neq \mathcal{A}_0(u)$, the maximum of $V(G, D_G^\star)$ is achieved if and only if $p_g = p_{data}$.*

*Proof.* When $p_g = p_{\text{data}}$, $D_G^\star(\boldsymbol{x}, u) = \frac{\mathcal{A}_1(u) + \mathcal{A}_0(u)}{2}$, we have:

$$V^\star(G, D_G^\star) = \int_u \mathcal{A}_1(u) \log \frac{2\mathcal{A}_1(u)}{\mathcal{A}_1(u) + \mathcal{A}_0(u)} + \mathcal{A}_0(u) \log \frac{2\mathcal{A}_0(u)}{\mathcal{A}_1(u) + \mathcal{A}_0(u)} du. \tag{11}$$

Subtracting $V^\star(G, D_G^\star)$ from $V(G, D_G^\star)$ gives:

$$V'(G, D_G^\star) = V(G, D_G^\star) - V^\star(G, D_G^\star)$$
$$= \int_{\boldsymbol{x}} \int_u (p_{\text{data}}(\boldsymbol{x})\mathcal{A}_1(u) + p_g(\boldsymbol{x})\mathcal{A}_0(u)) \log \frac{(p_{\text{data}}(\boldsymbol{x}) + p_g(\boldsymbol{x}))(\mathcal{A}_1(u) + \mathcal{A}_0(u))}{2(p_{\text{data}}(\boldsymbol{x})\mathcal{A}_1(u) + p_g(\boldsymbol{x})\mathcal{A}_0(u))} du dx, \tag{12}$$

$$= -2 \int_{\boldsymbol{x}} \int_u \frac{p_{\text{data}}(\boldsymbol{x})\mathcal{A}_1(u) + p_g(\boldsymbol{x})\mathcal{A}_0(u)}{2} \log \frac{\frac{p_{\text{data}}(\boldsymbol{x})\mathcal{A}_1(u) + p_g(\boldsymbol{x})\mathcal{A}_0(u)}{2}}{\frac{(p_{\text{data}}(\boldsymbol{x}) + p_g(\boldsymbol{x}))(\mathcal{A}_1(u) + \mathcal{A}_0(u))}{4}} du dx, \tag{13}$$

$$= -2\mathcal{D}_{\text{KL}} \left( \frac{p_{\text{data}}\mathcal{A}_1 + p_g\mathcal{A}_0}{2} \| \frac{(p_{\text{data}} + p_g)(\mathcal{A}_1 + \mathcal{A}_0)}{4} \right). \tag{14}$$

Since $V^\star(G, D_G^\star)$ is a constant with respect to $G$, maximizing $V(G, D_G^\star)$ is equivalent to maximizing $V'(G, D_G^\star)$. The optimal $V'(G, D_G^\star)$ is achieved if and only if the KL divergence reaches its minimum, where:

$$\frac{p_{\text{data}}\mathcal{A}_1 + p_g\mathcal{A}_0}{2} = \frac{(p_{\text{data}} + p_g)(\mathcal{A}_1 + \mathcal{A}_0)}{4}, \tag{15}$$

$$(p_{\text{data}} - p_g)(\mathcal{A}_1 - \mathcal{A}_0) = 0, \tag{16}$$

for any valid $\boldsymbol{x}$ and $u$. Hence, as long as there exists a valid $u$ that $\mathcal{A}_1(u) \neq \mathcal{A}_0(u)$, we have $p_{\text{data}} = p_g$ for any valid $\boldsymbol{x}$. $\qquad\qquad\square$

## 2.3 DISCUSSION

The theoretical analysis gives us more insights on RealnessGAN.

**Number of outcomes:** according to equation 16, each $u \in \Omega$ with $\mathcal{A}_0(u) \neq \mathcal{A}_1(u)$ may work as a constraint, pushing $p_g$ towards $p_{\text{data}}$. In the case of discrete distributions, along with the increment of the number of outcomes, the constraints imposed on $G$ accordingly become more rigorous and can cost $G$ more effort to learn. This is due to the fact that having more outcomes suggests a more fine-grained shape of the realness distribution for $G$ to match. In Sec.4, we verified that it is beneficial to update $G$ an increasing number of times before $D$'s update as the number of outcomes grows.

**Effectiveness of anchors:** view equation 16 as a cost function to minimize, when $p_{\text{data}} \neq p_g$, for some $u \in \Omega$, the larger the difference between $\mathcal{A}_1(u)$ and $\mathcal{A}_0(u)$ is, the stronger the constraint on $G$ becomes. Intuitively, RealnessGAN can be more efficiently trained if we choose $\mathcal{A}_0$ and $\mathcal{A}_1$ to be adequately different.

**Objective of $G$:** according to equation 3, the best way to fool $D$ is to increase the KL divergence between $D(\boldsymbol{x})$ and the anchor distribution $\mathcal{A}_0$ of fake samples, rather than decreasing the KL divergence between $D(\boldsymbol{x})$ and the anchor distribution $\mathcal{A}_1$ of real samples. It's worth noting that these two objectives are equivalent in the original work (Goodfellow et al., 2014). An intuitive explanation is that, in the distributional view of realness, realness distributions of real samples are not necessarily identical. It is possible that each of them corresponds to a distinct one. While $\mathcal{A}_1$ only serves as an anchor, it is ineffective to drag all generated samples towards the same target.

**Flexibility of RealnessGAN:** as a generalization of the standard framework, it is straightforward to integrate RealnessGAN with different GAN architectures, such as progressive GANs (Karras et al., 2018; 2019) and conditional GANs (Zhu et al., 2017; Ledig et al., 2017). Moreover, one may also combine the perspective of RealnessGAN with other reformulations of the standard GAN, such as replacing the KL divergence in equation 3 with the Earth Mover's Distance.

## 2.4 IMPLEMENTATION

In our implementation, the realness distribution $p_{\text{realness}}$ is characterized as a discrete distribution over $N$ outcomes $\Omega = \{u_0, u_1, ..., u_{N-1}\}$. Given an input sample $\boldsymbol{x}$, the discriminator $D$ returns $N$ probabilities on these outcomes, following:

$$p_{\text{realness}}(\boldsymbol{x}, u_i) = \frac{e^{\psi_i(\boldsymbol{x})}}{\sum_j e^{\psi_j(\boldsymbol{x})}}, \tag{17}$$

where $\psi = (\psi_0, \psi_1, ..., \psi_{N-1})$ are the parameters of $D$. Similarly, $\mathcal{A}_1$ and $\mathcal{A}_0$ are discrete distributions defined on $\Omega$.

As shown in the theoretical analysis, the ideal objective for $G$ is maximizing the KL divergence between $D(\boldsymbol{x})$ of generated samples and $\mathcal{A}_0$:

$$(G_{\text{objective1}}) \quad \min_G -\mathbb{E}_{\boldsymbol{z} \sim p_{\boldsymbol{z}}}[\mathcal{D}_{\text{KL}}(\mathcal{A}_0 \| D(G(\boldsymbol{z})))]. \tag{18}$$

However, as the discriminator $D$ is not always at its optimal, especially in the early stage, directly applying this objective in practice could only lead to a generator with limited generative power. Consequently, a regularizer is needed to improve $G$. There are several choices for the regularizer, such as the relativistic term introduced in (Jolicoeur-Martineau, 2019) that minimizes the KL divergence

between $D(\boldsymbol{x})$ of generated samples and random real samples, or the term that minimizes the KL divergence between $\mathcal{A}_1$ and $D(\boldsymbol{x})$ of generated samples, each of which leads to a different objective:

$$(G_{\text{objective2}}) \quad \min_G \quad \mathbb{E}_{\boldsymbol{x} \sim p_{\text{data}}, \boldsymbol{z} \sim p_{\boldsymbol{z}}}[\mathcal{D}_{\text{KL}}(D(\boldsymbol{x}) \| D(G(\boldsymbol{z})))] - \mathbb{E}_{\boldsymbol{z} \sim p_{\boldsymbol{z}}}[\mathcal{D}_{\text{KL}}(\mathcal{A}_0 \| D(G(\boldsymbol{z})))], \quad (19)$$

$$(G_{\text{objective3}}) \quad \min_G \quad \mathbb{E}_{\boldsymbol{z} \sim p_{\boldsymbol{z}}}[\mathcal{D}_{\text{KL}}(\mathcal{A}_1 \| D(G(\boldsymbol{z})))] - \mathbb{E}_{\boldsymbol{z} \sim p_{\boldsymbol{z}}}[\mathcal{D}_{\text{KL}}(\mathcal{A}_0 \| D(G(\boldsymbol{z})))]. \quad (20)$$

In Sec.4, these objectives are compared. And the objective in equation 19 is adopted as the default choice.

**Feature resampling.** In practice, especially in the context of images, we are learning from a limited number of discrete samples coming from a continuous data manifold. We may encounter issues caused by insufficient data coverage during the training process. Inspired by conditioning augmentation mentioned in (Zhang et al., 2016), we introduce a resampling technique performed on the realness output to augment data variance. Given a mini-batch $\{\boldsymbol{x}_0, ..., \boldsymbol{x}_{M-1}\}$ of size $M$, a Gaussian distribution $\mathcal{N}(\mu_i, \sigma_i)$ is fitted on $\{\boldsymbol{\psi}_i(\boldsymbol{x}_0), \boldsymbol{\psi}_i(\boldsymbol{x}_1), ..., \boldsymbol{\psi}_i(\boldsymbol{x}_{M-1})\}$, which are logits computed by $D$ on $i$-th outcome. We then resample $M$ new logits $\{\boldsymbol{\psi}'_i(\boldsymbol{x}_0), ..., \boldsymbol{\psi}'_i(\boldsymbol{x}_{M-1}); \boldsymbol{\psi}'_i \sim \mathcal{N}(\mu_i, \sigma_i)\}$ for $i$-th outcome and use them succeedingly.

The randomness introduced by resampling benefits the training of RealnessGAN in two aspects. First of all, it augments data by probing instances around the limited training samples, leading to more robust models. Secondly, the resampling approach implicitly demands instances of $\boldsymbol{\psi}_i(\boldsymbol{x})$ to be homologous throughout the mini-batch, such that each outcome reflects realness consistently across samples. We empirically found the learning curve of RealnessGAN is more stable if feature resampling is utilized, especially in the latter stage, where models are prone to overfit.

## 3 RELATED WORK

Generative adversarial network (GAN) was first proposed in (Goodfellow et al., 2014), which jointly learns a discriminator $D$ and a generator $G$ in an adversarial manner. Due to its outstanding learning ability, GANs have been adopted in various generative tasks (Radford et al., 2015; Yu et al., 2017; Zhu et al., 2017), among which Deep Convolutional GAN (DCGAN) (Radford et al., 2015) has shown promising results in image generation.

Although remarkable progress has been made. GAN is known to suffer from gradient diminishing and mode collapse. Variants of GAN have been proposed targeting these issues. Specifically, Wasserstein GAN (WGAN) Arjovsky et al. (2017) replaces JS-divergence with Earth-Mover's Distance, and Least-Square GAN (LSGAN) (Mao et al., 2017) transforms the objective of $G$ to Pearson divergence. Energy-based GAN (EBGAN) (Zhao et al., 2017) and Boundary Equilibrium GAN (BE-GAN) (Berthelot et al., 2017) employ a pre-trained auto-encoder as the discriminator, learning to distinguish between real and generated samples via reconstruction. Besides adjusting the objective of GAN, alternative approaches include more sophisticated architectures and training paradigms. Generally, ProgressiveGAN (Karras et al., 2018) and StyleGAN (Karras et al., 2019) propose a progressive paradigm, which starts from a shallow model focusing on a low resolution, and gradually grows into a deeper model to incorporate more details as resolution grows. On the other hand, COCO-GAN (Lin et al., 2019) tackles high resolution image generation in a divide-and-conquer strategy. It learns to produce decent patches at corresponding sub-regions, and splices the patches to produce a higher resolution image.

It's worth noting that many works on generative adversarial networks have discussed 'distributions' (Goodfellow et al., 2014; Radford et al., 2015; Arjovsky et al., 2017), which usually refers to the underlying distribution of samples. Some of the existing works aim to improve the original objective using different metrics to measure the divergence between the learned distribution $p_g$ and the real distribution $p_{\text{data}}$. Nevertheless, a single scalar is constantly adopted to represent the concept of realness. In this paper, we propose a complementary modification that models realness as a random variable follows the distribution $p_{\text{realness}}$. In the future work, we may study the combination of realness discriminator and other GAN variants to enhance the effectiveness and stability of adversarial learning.

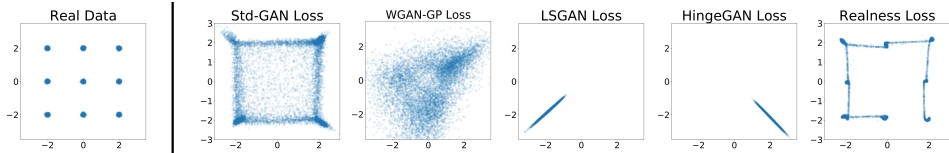

Figure 2: Left: real data sampled from the mixture of 9 Gaussian distributions. Right: samples generated by *Std-GAN*, *WGAN-GP*, *LSGAN*, *HingeGAN* and *RealnessGAN*.

## 4 EXPERIMENTS

In this section we study RealnessGAN from multiple aspects. Specifically, 1) we firstly focus on RealnessGAN's mode coverage ability on a synthetic dataset. 2) Then we evaluate RealnessGAN on CIFAR10 (32*32) (Krizhevsky, 2009) and CelebA (256*256) (Liu et al., 2015) datasets qualitatively and quantitatively. 3) Finally we explore RealnessGAN on high-resolution image generation task, which is known to be challenging for unconditional non-progressive architectures. Surprisingly, on the FFHQ dataset (Karras et al., 2019), RealnessGAN managed to generate images at the 1024*1024 resolution based on a non-progressive architecture. We compare *RealnessGAN* to other popular objectives in generative adversarial learning, including the standard GAN (*Std-GAN*) (Radford et al., 2015), *WGAN-GP* (Arjovsky et al., 2017), *HingeGAN* (Zhao et al., 2017) and *LSGAN* (Mao et al., 2017).

For experiments on synthetic dataset, we use a generator with four fully-connected hidden layers, each of which has $400$ units, followed by batch normalization and ReLU activation. The discriminator has three fully-connected hidden layers, with $200$ units each layer. LinearMaxout with $5$ maxout pieces are adopted and no batch normalization is used in the discriminator. The latent input $z$ is a $32$-dimensional vector sampled from a Gaussian distribution $\mathcal{N}(\mathbf{0}, \boldsymbol{I})$. All models are trained using Adam (Kingma & Ba, 2015) for $500$ iterations.

On real-world datasets, the network architecture is identical to the DCGAN architecture in Radford et al. (2015), with the prior $p_z(z)$ a 128-dimensional Gaussian distribution $\mathcal{N}(\mathbf{0}, \boldsymbol{I})$. Models are trained using Adam (Kingma & Ba, 2015) for $520k$ iterations. To guarantee training stability, we adopt settings that are proved to be effective for baseline methods. Batch normalization (Ioffe & Szegedy, 2015) is used in $G$, and spectral normalization (Miyato et al., 2018) is used in $D$. For WGAN-GP we use $lr = 1e - 4, \beta_1 = 0.5, \beta_2 = 0.9$, updating $D$ for 5 times per $G$'s update (Gulrajani et al., 2017); for the remaining models, we use $lr = 2e - 4, \beta_1 = 0.5, \beta_2 = 0.999$, updating $D$ for one time per $G$'s update (Radford et al., 2015). Fréchet Inception Distance (FID) (Heusel et al., 2017) and Sliced Wasserstein Distance (SWD) (Karras et al., 2018) are reported as the evaluation metrics. Unless otherwise stated, $\mathcal{A}_1$ and $\mathcal{A}_0$ are chosen to resemble the shapes of two normal distributions with a positive skewness and a negative skewness, respectively. In particular, the number of outcomes are empirically set to 51 for CelebA and FFHQ datasets, and 3 for CIFAR10 dataset.

### 4.1 SYNTHETIC DATASET

Since $p_{\text{data}}$ is usually intractable on real datasets, we use a toy dataset to compare the learned distribution $p_g$ and the data distribution $p_{\text{data}}$. The toy dataset consists of $100,000$ 2D points sampled from a mixture of 9 isotropic Gaussian distributions whose means are arranged in a 3 by 3 grid, with variances equal to $0.05$. As shown in Fig.2, the data distribution $p_{\text{data}}$ contains 9 welly separated modes, making it a difficult task despite its low-dimensional nature.

To evaluate $p_g$, we draw $10,000$ samples and measure their quality and diversity. As suggested in (Dumoulin et al., 2016), we regard a sample as of high quality if it is within $4\sigma$ from the $\mu$ of its nearest Gaussian. When a Gaussian is assigned with more than $100$ high quality samples, we consider this mode of $p_{\text{data}}$ is recovered in $p_g$. Fig.2 visualizes the sampled points of different methods, where *LSGAN* and *HingeGAN* suffer from significant mode collapse, recovering only a single mode. Points sampled by *WGAN-GP* are overly disperse, and only $0.03\%$ of them are of high quality. While *Std-GAN* recovers 4 modes in $p_{\text{data}}$ with $32.4\%$ high quality samples, 8 modes are recovered by *RealnessGAN* with $60.2\%$ high quality samples. The average $\sigma$s of these high quality samples in *Std-GAN* and *RealnessGAN* are respectively $0.083$ and $0.043$. The results suggest that treating realness as a random variable rather than a single scalar leads to a more strict discriminator

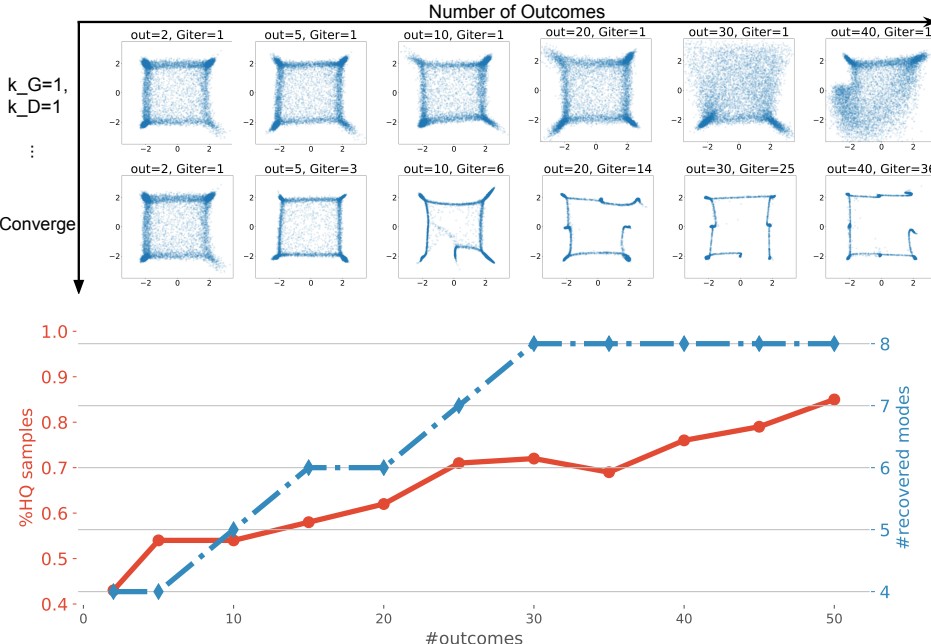

Figure 3: First row: the results of *RealnessGAN* when fixing $k_G = k_D = 1$ and increasing the number of outcomes. Second row: the results of *RealnessGAN* when $k_G$ is properly increased. Bottom curves: under the settings of second row, the ratio of high quality samples and the number of recovered modes.

that criticizes generated samples from various aspects, which provides more informative guidance. Consequently, $p_g$ learned by *RealnessGAN* is more diverse and compact.

We further study the effect of adjusting the number of outcomes in the realness distribution $p_{\text{realness}}$ on this dataset. To start with, we fix $k_G$ and $k_D$ to be 1, which are the number of updates for $G$ and $D$ in one iteration, and adjust the number of outcomes of $p_{\text{realness}}$, $\mathcal{A}_0$ and $\mathcal{A}_1$. As shown in the first row of Fig.3, it can be observed that in general $G$ recovers less modes as the number of outcomes grows, which is a direct result of $D$ becoming increasingly rigorous and imposing more constraints on $G$. An intuitive solution is to increase $k_G$ such that $G$ is able to catch up with current $D$. The second row of Fig.3 demonstrates the converged cases achieved with suitable $k_G$s, suggesting *RealnessGAN* is effective when sufficient learning capacity is granted to $G$. The ratio of high quality samples $r_{\text{HQ}}$ and the number of recovered modes $n_{\text{mode}}$ in these cases are plotted in Fig.3. The two curves imply that besides $k_G$, $r_{\text{HQ}}$ and $n_{\text{mode}}$ are all positively related to the number of outcomes, validating that measuring realness from more aspects leads to a better generator.

## 4.2 REAL-WORLD DATASETS

As GAN has shown promising results when modeling complex data such as natural images, we evaluate *RealnessGAN* on real-world datasets, namely CIFAR10, CelebA and FFHQ, which respectively contains images at 32*32, 256*256 and 1024*1024 resolutions. The training curves of baseline methods and *RealnessGAN* on CelebA and CIFAR10 are shown in Fig.4. The qualitative results measured in FID and SWD are listed in Tab.1. We report the minimum, the maximum, the mean and the standard deviation computed along the training process. On both datasets, compared to baselines, *RealnessGAN* obtains better scores in both metrics. Meantime, the learning process of *RealnessGAN* is smoother and steadier (see SD in Tab.1 and curves in Fig.4). Samples of generated images on both datasets are included in Fig.8.

On FFHQ, we push the resolution of generated images to 1024*1024, which is known to be challenging especially for a non-progressive architecture. As shown in Fig.8, despite building on a relatively simple DCGAN architecture, *RealnessGAN* is able to produce realistic samples from scratch at such a high resolution. Quantitatively, *RealnessGAN* obtains an FID score of 17.18. For reference, our

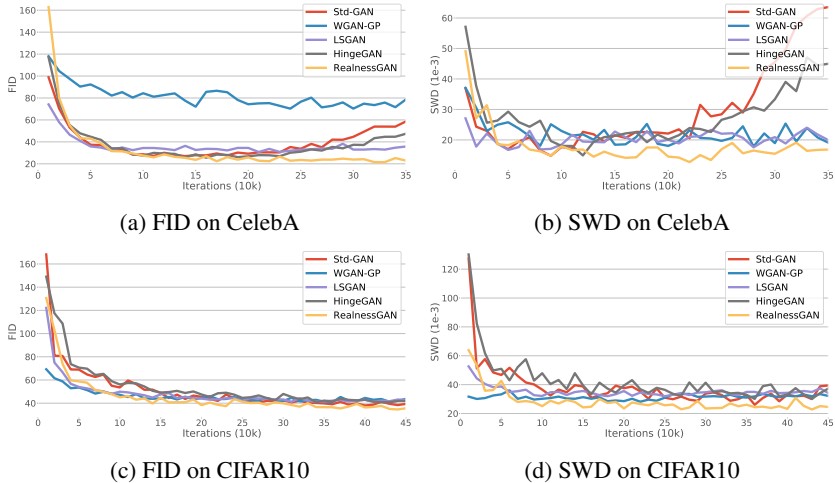

(a) FID on CelebA          (b) SWD on CelebA

(c) FID on CIFAR10          (d) SWD on CIFAR10

Figure 4: Training curves of different methods in terms of FID and SWD on both CelebA and CIFAR10, where the raise of curves in the later stage indicate mode collapse. Best viewed in color.

Table 1: Minimum (min), maximum (max), mean and standard deviation (SD) of FID and SWD on CelebA and CIFAR10, calculated at 20k, 30k, ... iterations. The best indicators in baseline methods are underlined.

|  | **Method** | **FID ↓** | | | | **SWD ($\times 10^3$) ↓** | | | |
|---|---|---|---|---|---|---|---|---|---|
|  |  | Min | Max | Mean | SD | Min | Max | Mean | SD |
| **CelebA** | Std-GAN | 27.02 | 70.43 | 34.85 | 9.40 | 14.81 | 68.06 | 30.58 | 15.39 |
|  | WGAN-GP | 70.28 | 104.60 | 81.15 | 8.27 | 17.85 | 30.56 | 22.09 | 2.93 |
|  | LSGAN | 30.76 | 57.97 | 34.99 | 5.15 | 16.72 | 23.99 | 20.39 | 2.25 |
|  | HingeGAN | 25.57 | 75.03 | 33.89 | 10.61 | 14.91 | 54.30 | 28.86 | 10.34 |
|  | RealnessGAN | **23.51** | 81.3 | **30.82** | 7.61 | **12.72** | 31.39 | **17.11** | 3.59 |
| **CIFAR10** | Std-GAN | 38.56 | 88.68 | 47.46 | 15.96 | 28.76 | 57.71 | 37.55 | 7.02 |
|  | WGAN-GP | 41.86 | 79.25 | 46.96 | 5.57 | 28.17 | 36.04 | 30.98 | 1.78 |
|  | LSGAN | 42.01 | 75.06 | 48.41 | 7.72 | 31.99 | 40.46 | 34.75 | 2.34 |
|  | HingeGAN | 42.40 | 117.49 | 57.30 | 20.69 | 32.18 | 61.74 | 41.85 | 7.31 |
|  | RealnessGAN | **34.59** | 102.98 | **42.30** | 11.84 | **22.80** | 53.38 | **26.98** | 5.47 |

re-implemented *StyleGAN* (Karras et al., 2019) trained under a similar setting receives an FID score of 16.12. These results strongly support the effectiveness of *RealnessGAN*, as *StyleGAN* is one of the most advanced GAN architectures so far.

## 4.3 ABLATION STUDY

The implementation of *RealnessGAN* offers several choices that also worth digging into. On synthetic dataset, we explored the relationship between the number of outcomes and $G$'s update frequency. On real-world dataset, apart from evaluating *RealnessGAN* as a whole, we also studied the affect of feature resampling, different settings of $\mathcal{A}_0$ and $\mathcal{A}_1$ and choices of $G$'s objective.

Table 2: Minimum (min), maximum (max), mean and standard deviation (SD) of FID on CelebA using different anchor distributions, calculated at 20k, 30k, ... iterations.

| $\mathcal{D}_{\mathrm{KL}}(\mathcal{A}_1\|\mathcal{A}_0)$ | Min | Max | Mean | SD |
|---|---|---|---|---|
| 1.66 | 31.01 | 96.11 | 40.75 | 11.83 |
| 5.11 | 26.22 | 87.98 | 36.11 | 9.83 |
| 7.81 | 25.98 | 85.51 | 36.30 | 10.04 |
| 11.05 | 23.51 | 81.30 | 30.82 | 7.61 |

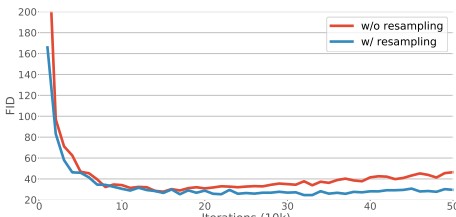

Figure 5: Training FID curves of *RealnessGAN* with and without feature re-sampling.

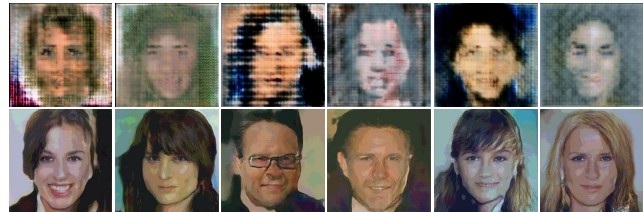

Figure 6: Samples generated by *RealnessGAN* trained with the ideal objective (equation 18). Top-row: samples when $\mathcal{D}_{\mathrm{KL}}(\mathcal{A}_1\|\mathcal{A}_0) = 11.05$. Bottom-row: samples when $\mathcal{D}_{\mathrm{KL}}(\mathcal{A}_1\|\mathcal{A}_0) = 33.88$.

Table 3: FID scores of $G$ on CIFAR10, trained with different objectives.

| G Objective | FID |
|---|---|
| Objective1 (equation 18) | 36.73 |
| Objective2 (equation 19) | 34.59 |
| Objective3 (equation 20) | 36.21 |
| DCGAN | 38.56 |
| WGAN-GP | 41.86 |
| LSGAN | 42.01 |
| HingeGAN | 42.40 |

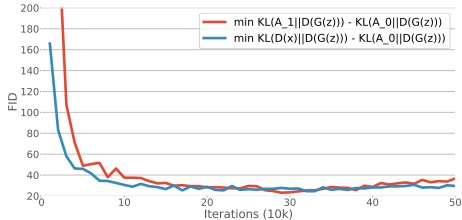

Figure 7: Training curves of *RealnessGAN* on CelebA using objective2 (equation 19) and objective3 (equation 20).

**Feature Resampling.** Fig.5 shows the training curves of *RealnessGAN* with and without feature resampling. It can be noticed that despite the results are similar, feature resampling stabilizes the training process especially in the latter stage.

**Effectiveness of Anchors.** Tab.2 reports the results of varying the KL divergence between anchor distributions $\mathcal{A}_0$ and $\mathcal{A}_1$. The FID score indicates that, as the KL divergence between $\mathcal{A}_0$ and $\mathcal{A}_1$ increases, *RealnessGAN* tends to perform better, which verifies our discussion in Sec.2.3 that a larger difference between anchor distributions imposes stronger constraints on $G$. To further testify, two different pairs of anchors with similar KL divergences (11.95 and 11.67) are exploited and they yield comparable FID scores (23.98 and 24.22).

**Objective of G.** As mentioned in Sec.2.3, theoretically, the objective of $G$ is $\max_G \mathbb{E}_{\boldsymbol{x}\sim p_g}[\mathcal{D}_{\mathrm{KL}}(\mathcal{A}_0\|D(\boldsymbol{x}))]$. However, in practice, since $D$ is not always optimal, we need either a pair of $\mathcal{A}_0$ and $\mathcal{A}_1$ that are drastically different, or an additional constraint to aid this objective. Fig.6 shows that, with the ideal objective alone, when the KL divergence between $\mathcal{A}_0$ and $\mathcal{A}_1$ is sufficiently large, on CelebA we could obtain a generator with limited generative power. On the other hand, by applying constraints as discussed in Sec.2.4, $G$ can learn to produce more realistic samples as demonstrated in Fig.8. Similar results are observed on CIFAR10, where *RealnessGAN* obtains comparable FID scores with and without constraints, as shown in Tab.3. Fig.7 also provides the training curves of *RealnessGAN* on CelebA using these two alternative objectives.

## 5 CONCLUSION

In this paper, we extend the view of realness in generative adversarial networks under a distributional perspective. In our proposed extension, RealnessGAN, we represent the concept of realness as a realness distribution rather than a single scalar. so that the corresponding discriminator estimates realness from multiple angles, providing more informative guidance to the generator. We prove RealnessGAN has theoretical guarantees on the optimality of the generator and the discriminator. On both synthetic and real-world datasets, RealnessGAN also demonstrates the ability of effectively and steadily capturing the underlying data distribution.

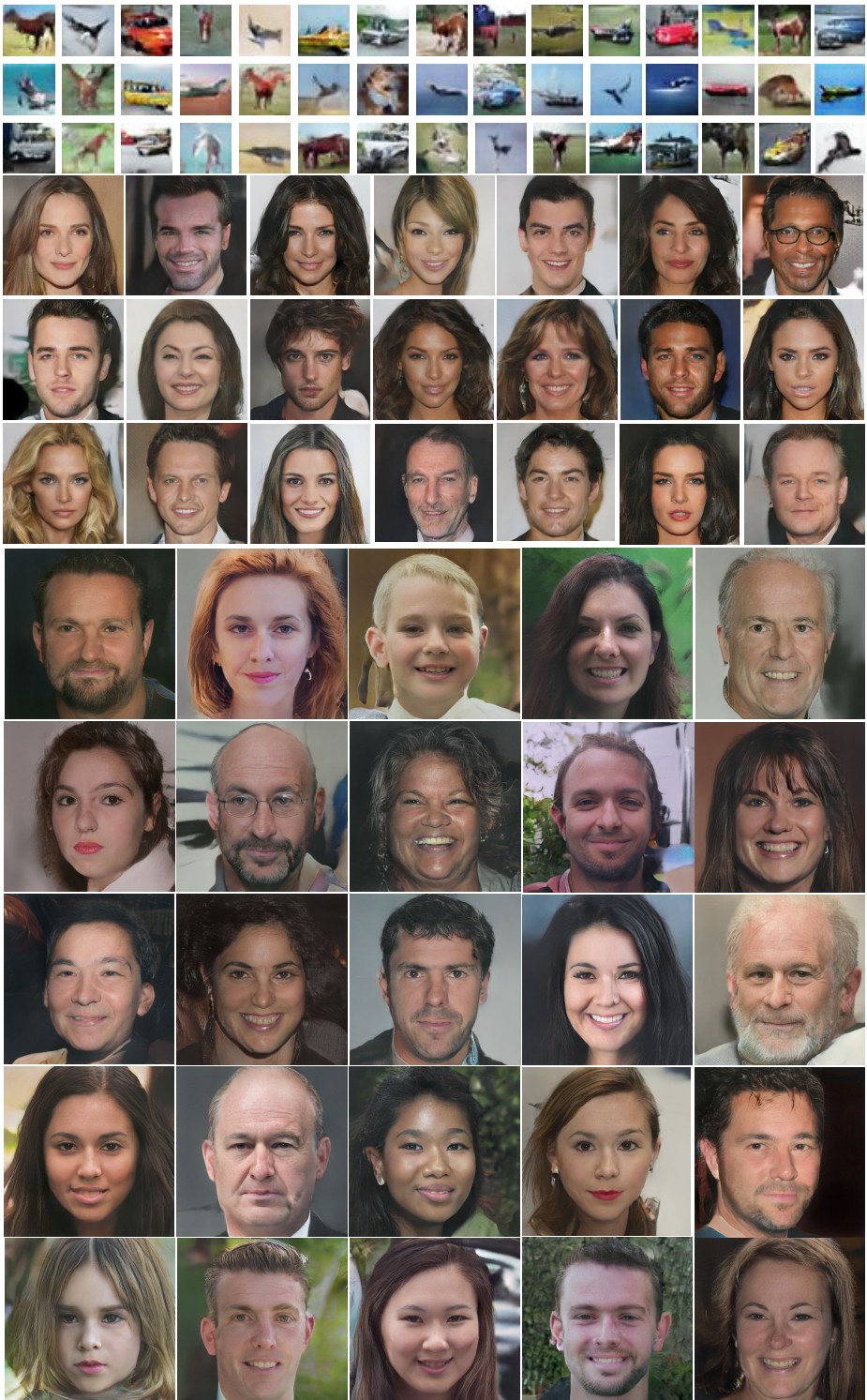

Figure 8: Images sampled from *RealnessGAN*, respectively trained on CIFAR10 (top), CelebA (middle) and FFHQ (bottom).

**Acknowledgement**  We thank Zhizhong Li for helpful discussion on the theoretical analysis. This work is partially supported by the Collaborative Research Grant of "Large-scale Multi-modality Analytics" from SenseTime (CUHK Agreement No. TS1712093), the General Research Funds (GRF) of Hong Kong (No. 14209217 and No. 14205719), Singapore MOE AcRF Tier 1, NTU SUG, and NTU NAP.

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
