# OpenReview forum: "Real or Not Real, that is the Question"
_ICLR.cc/2020/Conference — Accept (Spotlight)_

### Official Review · AnonReviewer3 · 2019-10-11
**Official Blind Review #3**

**Rating:** 6

**Review:**

Post rebuttal: The authors' responses have addressed most of my concerns, and I've raised my rating from 3 to 6.

----------------------------------------

Summary:
This paper extends the discriminator of GAN to use a distributional output (multiple scalars) instead of a single scalar. As a result, the trained GAN becomes robust to the mode collapse.

Pros:
- The proposed method is clearly written and well-justified (e.g., Theorem 2).
- Extension of the relativistic GAN [1] to the proposed setting is interesting.
- The authors demonstrate that vanilla DCGAN architecture can generate high-fidelity (1024x1024) images.

Cons:

1. An ensemble of discriminators?

The authors use multiple scalars to consider diverse factors of the realness. However, it is simply an ensemble of discriminators [2] in a spirit. As each discriminator focus on different factors, it is not surprising that the generator becomes robust to the mode collapse. Also, recent work on mode collapse (e.g., [3]) shows better results on the mixture of gaussian experiments even using a single discriminator. At least, the authors should compare their method with the ensemble methods and claim the advantage over them.

2. Choice of the anchor distributions.

The choice of anchor distributions A_0 and A_1 are not specified. While the authors provide some partial results in Table 2, it would be worthwhile to clarify the experimental details and justify them.

3. Role of each outcome u_i?

The authors claim that each outcome u_i corresponds to the different factors of realness. However, the role of learned u_i is not investigated. Also, one may enforce u_i to learn different factors by promoting diversity of them, e.g., decrease their cosine similarity [4].

Minor comments:
- The word "support" [5] is misused. The support itself means the set of non-zero elements, hence the authors should use the word "outcome" (or "sample") instead of "support".
- The notation is not consistent. For example, the authors may use "x \sim p_data(x)" (specify variable) or "z \sim p_z" (omit variable), but not both.
- Numbering is not consistent. For example, "Tab.4.2." should be changed to "Tab.2." for consistency.


[1] Jolicoeur-Martineau. The relativistic discriminator: a key element missing from standard GAN. ICLR 2019.
[2] Durugkar et al. Generative Multi-Adversarial Networks. ICLR 2017.
[3] Xiao et al. BourGAN: Generative Networks with Metric Embeddings. NeurIPS 2018.
[4] Elfeki et al. GDPP: Learning Diverse Generations Using Determinantal Point Process. ICML 2019.
[5] https://en.wikipedia.org/wiki/Support_(mathematics)

**Experience Assessment:**

I have published one or two papers in this area.

**Review Assessment: Checking Correctness Of Derivations And Theory:**

I carefully checked the derivations and theory.

**Review Assessment: Checking Correctness Of Experiments:**

I carefully checked the experiments.

**Review Assessment: Thoroughness In Paper Reading:**

I read the paper thoroughly.

---

> ### Author Response · Authors · 2019-11-12
> **Reply to Reviewer #3**
>
> Thank you for your comments. We address your concerns below and code will be released. Revision in the paper is highlighted in magenta.
>
> **Minor comments**
> Thank you for pointing them out. We have revised our paper to fix these errors.
>
> **An ensemble of discriminators**
> EnsembledGAN and RealnessGAN are significantly different both conceptually and technically.
>
> *Conceptually*:  EnsembledGAN aims at balancing n *independent* discriminators. They all treat realness as a scalar, ranging from 0 to 1. On the contrary, RealnessGAN treats realness as a random variable and use a *single* discriminator to capture the distribution of the variable. Through the distributional constraint, outcomes of RealnessGAN have semantic meanings, i.e. assessing images from multiple angles. Such semantics are implicitly constrained in our paper. And we could also deploy an explicit constraint to enforce this. Such an updated view on the realness lead to theoretical guarantees on the optimality.
>
> Conceptually RealnessGAN and EnsembledGAN are orthogonal. RealnessGAN could serve as one of the discriminators of EnsembledGAN by taking the expectation of estimated realness distribution as its output.
>
> *Technically*: EnsembleGAN uses multiple discriminators that could have different architectures and weights. RealnessGAN uses a single discriminator. While some of the resemblance comes from the fact that we use a discrete distribution to approximate the realness distribution, so that outcomes are discrete, RealnessGAN has the potential to use a continuous distribution to represent the realness distribution, which further distinguishes it from the EnsembledGAN. Moreover, unlike EnsembledGAN, the outcomes of RealnessGAN  have semantic meanings, we could also extend RealnessGAN to include structures (e.g. grids, 3d lattice, etc) for discrete outcomes or priors on continuous outcomes, further improving RealnessGAN.
>
> *Result*: Despite the conceptual differences, we compare RealnessGAN and EnsembledGAN on Cifar10, and the results are:
>                                  FID           SWD
> RealnessGAN       34.59         22.80
> EnsembledGAN   37.76         26.53
> where RealnessGAN is shown to outperform EnsembledGAN. EnsembledGAN using DCGAN also fails on FFHQ, which is more challenging.
>
> **BourGAN**
> The work of BourGAN is orthogonal to us. BourGAN provides a technique on the latent space of z to help the standard GAN. RealnessGAN replaces the estimation of realness with a distributional view. One could combine BourGAN and RealnessGAN.
>
> **Choice of the anchor distributions**
> In the experiments, we choose A_0 and A_1 to resemble the shape of normal distributions with a positive skew and a negative skew, respectively.
>
> In the paper, we include several clues on how to choose A_0 and A_1. 1) We show in the theoretical analysis A_0 and A_1 can be chosen flexibly, as long as they satisfy A_0(u) \ne A_1(u) for some outcome u. 2) We also show in Table 2, the KL divergence between A_0 and A_1 is important. 3) To show Table 2 is sufficient to guide how to choose A_0 and A_1, we extend the study in Table 2 to compare pairs of A_0 and A_1 that have different shapes but similar KL divergences, and the results are:
>                 KL	       FID(min)
> Pair1     11.95         29.62
> Pair2     11.67         30.12
> The results suggest that the major factor we need to care about is the KL divergence between our chosen A_0 and A_1.
>
> **Role of each outcome u_i**
> The role of outcomes are currently implicitly constrained by the distributional view during training. We include a heuristic interpretation on the estimated distributions in Appendix C, where we use the generator of RealnessGAN to produce a set of samples and the discriminator to produce their estimated realness distributions. Consequently, we have found generated samples that have similar realness distributions have something in common.
> Explicitly adding a regularizer to better disentangle the semantics into different outcomes is possible but we think it is beyond the focus of this paper, we thus leave it as the future work.
>
> **Extension of the relativistic GAN [1] to the proposed setting**
> We would like to clarify that the relativistic loss used here serves as a regularizer, so that the objective is:
> (objective 1)   min KL(D(x) || D(G(z))) - KL(A_0 || D(G(z)))
> We also replace it with an alternative one, so that the objective becomes:
> (objective 2)    min KL(A_1 || D(G(z))) - KL(A_0 || D(G(z)))
> The quantitative results of these objectives on Cifar10 are:
>                           FID
> Objective1     34.59
> Objective2     36.21
> DCGAN          38.56
> WGAN-GP     41.86
> LSGAN           42.01
> HingeGAN    42.40
> we have also included  in Appendix A the training curves of using objective 1 and objective 2 on CelebA. On both datasets, we can see without the relativistic loss, RealnessGAN can still outperform baselines. We have revised paper to clarify this ambiguity.

---

> > ### Comment · AnonReviewer3 · 2019-11-12
> > **Reply to the authors**
> >
> > Thank you for your response. Many parts of my concerns are resolved. However, the justification and the interpretation of the anchor distributions are still unclear.
> >
> > 1.  What is the motivation of using normal distributions with a positive skew and a negative skew? The authors provide some additional empirical results in the response, but the investigation is still unsatisfactory. Also, many details are yet omitted, e.g., what is the "exact" distributions of A_0 and A_1 (not just "resemble" normal), and what are the "different shapes" that compared with?
> >
> > 2. Why not the shape of anchor distributions, but the KL divergence is important? Are there some explanations? Also, Table 2 shows that lager KL divergence makes a better generation. Is it also true for even larger values? For example, the authors may extrapolate larger KL divergences (say 100) and report if the trend is consistent.
> >
> > 3. I appreciate the authors provide some interpretations in Appendix C. However, the provided figures lack some details, e.g., how two samples (that has similar output distributions) are selected. Also, the interpretation of the shape of the plots is not discussed.
> >
> > Overall, I think the paper is interesting, but it lacks some important details and the discussion on the design choices are not fully studied. If these concerns are resolved, I am willing to raise the score.

---

> > > ### Author Response · Authors · 2019-11-13
> > > **Reply to Reviewer #3**
> > >
> > > **Why KL divergence important**
> > > At first, we would like to clarify that as the shapes of anchor distributions are correlated to their KL divergence. The shapes of anchor distributions are factors, but the major factor is their KL divergence.
> > >
> > > In other words, there are different pairs of distributions that could have similar KL divergence. Naturally one may guess even with a similar KL divergence, would the actual shapes (this term may not be mathematically correct) of distributions affect the results. We thus investigate which one is the major factor.
> > >
> > > Please read the theoretical analysis especially theorem 2, and the discussion on the number of effective outcomes, as they are very important to show why KL divergence is important.
> > >
> > > Specifically, our explanation comes from the following aspects:
> > >
> > > 1.Theorem 2 indicates as long as there are A0(u) \ne A1(u) for some outcome u, the optimal G satisfies p_g = p_data when D is optimal. This observation intuitively suggests A0 and A1 should be different.
> > >
> > > 2.In the discussion, we guess the number of outcomes in A0 and A1 is important. Because intuitively with more number of outcomes, you are more likely to get different A0 and A1. This hypothesis is supported by the experiment results in Figure 3.
> > >
> > > 3.And in the discussion, we also guess the effectiveness of each outcome u is associated with the difference between A0(u) and A1(u). Intuitively, when you view the equation 12 as an objective for p_g (i.e. J(p_g) = |A0(u) - A1(u)| * |p_g - p_data|), it indicates larger the |A0(u) - A1(u)| is, stronger the constraint that pushes p_g towards p_data.
> > >
> > > 4.From the above aspects, we could hypothesis the difference between A0 and A1 is very important. And one of the standard metrics that measure differences between distributions is KL divergence.
> > >
> > > As shown in Table 2 and our first response, it’s indeed a major factor for the quality of trained G.
> > >
> > > **KL divergence further increases**
> > > Thanks for suggesting what will happen when we further increase the KL divergence between A0 and A1 to a very large value. Although we do not conduct such studies, we could gain some intuitions from the success of the standard GAN.
> > >
> > > Specifically, please refer to the beginning of sec2.2 where we find the standard GAN can be seen as a special case of RealnessGAN with *2* outcomes, A0 = [1, 0], and A1 = [0,1]. While the KL divergence between [0,1] and [1,0] is infinity, the standard GAN still could produce good generators. We thus hypothesize that when further increasing the KL divergence between A0 and A1, performance saturation or overfitting (Figure 3) may happen, but the objective of RealnessGAN will not lead to total collapse.
> > >
> > > **Motivation of the choice of normal distributions**
> > > At first please refer to https://en.wikipedia.org/wiki/Skewness for the shapes of A0 and A1.
> > >
> > > And such a hyperparameter is chosen by 1) our investigation on which factor, namely the shape and the difference, is the major factor, 2) validation on an isolated subset.
> > >
> > > We use the term “resemble” because we are approximating normal distributions using discrete distributions, which looks like histograms. Sorry for not include the number of outcomes. We use 51 outcomes and 3 outcomes respectively on CelebA and Cifar10. With the number of outcomes, you could then get A0 and A1 accordingly.
> > >
> > > Sorry for the ambiguity in the meaning of different shapes. We mean there are various pairs of different distributions that could have a similar KL divergence. And one would naturally guess under the assumption they have a similar KL divergence, whether the choice on a specific pair of distributions affects the final results significantly. We thus conduct a quick study where we use two different pairs of A0 and A1 that have a similar KL divergence.
> > >
> > > **Interpretation in Appendix C**
> > > The samples in Appendix C are *heuristically* sampled by us. The main purpose of Appendix C is to show that estimated realness distribution could potentially reflect the distributional nature of realness, so that we could explore along this direction in the future. The heuristic results in Appendix C briefly show that samples with a similar realness distribution have something in common. We do not include further interpretations on the semantics of each outcome because this is not the focus of our work.

---

> > > > ### Comment · AnonReviewer3 · 2019-11-13
> > > > **Reply to the authors**
> > > >
> > > > Thank you for your response. Most of my concerns are addressed, and I've raised my rating from 3 to 6.
> > > >
> > > > Nevertheless, I suggest the authors
> > > > 1) clarify the writing and specify the details (e.g., specify the outcomes for CelebA and CIFAR-10 are 51 and 3, respectively)
> > > > 2) add more ablation studies to justify their choices (e.g., even though the standard GAN works pretty well while having the KL divergence between A0 = [1, 0] and A1 = [0,1] of infinity, it is worse than the proposed RealnessGAN. Hence, there may be a sweet spot in some finite KL divergence)
> > > > in the next revised paper to further strengthen this work.

---

> > > > > ### Author Response · Authors · 2019-11-15
> > > > > **Reply to Reviewer #3**
> > > > >
> > > > > Thank you for your timely and valuable feedback, and we are glad to know our rebuttal could address your concerns.
> > > > >
> > > > > As for your suggestions:
> > > > >
> > > > > 1.Sure. We will update our paper accordingly and release the code that ensures the reproducibility of our results, because we believe this paper provides several interesting directions for future work, both theoretically and technically. Releasing our code could bring interested people in the game quickly.
> > > > >
> > > > > We include an illustrative figure in Appendix D for our choice of A0 and A1. We do not write a lot about our choice in the paper for we do not want to leave an impression that only the precise re-choice of our chosen A0 and A1 could lead to the results reported in the paper. In fact, we didn’t spend much effort on choosing A0 and A1. In our experience, we care more about the KL divergence between A0 and A1, for we found both theoretically and empirically it is important. And we only adjust their shapes when we feel it’s hard to find a specific pair that has sufficient KL divergence using current shapes.
> > > > >
> > > > > 2. The main difficulty of this study is that it's really hard to find two reasonable distributions that have an extremely large KL divergence, especially when you fix other hyperparameters (the number of outcomes, etc.) Here the term reasonable means we do not want to use strange discrete distributions (e.g. one has zeros on many outcomes), which may bring unexpected noise.
> > > > >
> > > > > Nevertheless, due to the time limit, we have conducted one additional study for Table 2, where we further increased the KL divergence between A0 and A1 to 36.84. In this case, although we didn’t sufficiently train the RealnessGAN because of time, we obtain a FID score of 26.41. It seems the performance will saturate. We will re-check this after sufficient training is applied in this case.

---

### Official Review · AnonReviewer1 · 2019-10-24
**Official Blind Review #1**

**Rating:** 6

**Review:**

Update: I raised my score from 3 to 6 after the authors addressed most of my comments.

====================================================

This paper propose a new GAN formulation where the Discriminator outputs a discrete probability distribution instead of a scalar for each inputs. This discrete probability distribution outputted by the discriminator is then compared, using the KL divergence, to two different reference distributions according to if the image is from the dataset or generated. The paper show that the proposed approach is a generalization of the standard GAN. They then prove that under some condition that similarly to GAN at the optimum $p_g = p_data$. In addition the paper propose two tricks 1) they include an additional term in the loss for the generator, such that the generator is also trying to minimize the KL between the discriminator distribution for a generated and the discriminator distribution for a real samples.  2) They propose some procedure to resample the logits of the Discriminator.
They show on a toy example how increasing the dimension of the distribution of the Discriminator also increases the performance. They also show that their method can slightly improve performance on CelebA and CIFAR10 and that it can also scale to high resolution images on FFHQ.

I'm in favour of rejecting this paper. In particular I find the method not very well motivated for several reasons. First it's never explained in the text how the reference distribution $A_0$ and $A_1$ are chosen, and so it's not clear what they represent. Second it's not clear why we need the two tricks or wether the method would also work without the proposed tricks.

Main Argument:
- Please provide an explanation how $A_0$ and $A_1$ are chosen and what are the $A_0$ and $A_1$ chosen in the experiments. This seems very critical to the performance of the method as shown in table 2 but explained nowhere in the paper.
- What is the effect of the "Relativistic" loss ? Does the method work if we remove it ? An extended ablation study would be nice and give more insight on what is really important for the performance of the method ?
- The author claim in the experimental section that their method perform betters on both datasets, however when looking at table 1, WGAN-GP performs better on CIFAR10 and when looking at the standard deviation we can see that the improvment is not significative.


**Experience Assessment:**

I have published in this field for several years.

**Review Assessment: Checking Correctness Of Derivations And Theory:**

I assessed the sensibility of the derivations and theory.

**Review Assessment: Checking Correctness Of Experiments:**

I carefully checked the experiments.

**Review Assessment: Thoroughness In Paper Reading:**

I read the paper at least twice and used my best judgement in assessing the paper.

---

> ### Author Response · Authors · 2019-11-12
> **Reply to Reviewer #1**
>
> Thank you for your comments. Below we address the concerns. Code to reproduce our results will be released. We also include our revision in the paper (content in magenta).
>
> **What are A0 and A1**
> At first we have shown in the introduction that we treat realness as a random variable, when assessing from different angles, we could obtain different scores. Consequently, we use the discriminator to estimate a realness distribution.
>
> The training of discriminator requires two virtual “ground-truths” or anchors, which represent the realness of real and fake images, respectively. In the scalar case (i.e. the standard GAN), 0 and 1 are chosen. And it’s also possible to replace 0 and 1 with other scalars, such as -1 and 1, etc. Similar to the standard GAN, we thus choose two anchoring distribution A_0 and A_1 to train RealnessGAN. A_0 and A_1 represent the virtual “ground-truths” real distributions of real and fake images.
>
> **How are A_0 and A_1 chosen**
> In the experiments, we choose A_0 and A_1 to resemble the shapes of two normal distributions with a positive skew and a negative skew, respectively.
> And such a hyperparameter is chosen by 1) our investigation on which factor, namely the shape and the difference, is the major factor, 2) validation on an isolated subset.
> We use the term “resemble” because we are approximating normal distributions using discrete distributions, which looks like histograms. Sorry for not include the number of outcomes. We use 51 outcomes and 3 outcomes respectively on CelebA and Cifar10. With the number of outcomes, you could then get A0 and A1 accordingly.
>
> In the paper we include several clues on how to choose A_0 and A_1. 1) We show in the theoretical analysis A_0 and A_1 can be chosen flexibly, as long as they satisfy A_0(u) \ne A_1(u) for some outcome u. 2) We also show in Table 2, the KL divergence between A_0 and A_1 is important. 3) To further show that Table 2 is sufficient to guide the selection of A_0 and A_1, we extend the study in Table 2 to compare two different pairs of A_0 and A_1 that have similar KL divergences, and the results on CelebA are:
>                 KL	            FID
>  Pair1     11.95         23.98
>  Pair2     11.67         24.22
> The results suggest that the major factor we need to care about is the KL divergence between the chosen A_0 and A_1.
>
> ** Feature resampling **
> The effect of feature re-sampling is studied in Figure.6, where RealnessGAN could obtain similar results without this technique.
>
> ** The effect to relativistic loss **
> The relativistic loss is used as a regularizer. Specifically, we show in the theoretical analysis the ideal objective of G is:
> (objective 1)   min -KL(A_0 || D(G(z)))
> we show in the Appendix A such an overly-loosen objective could indeed lead to an acceptable generator. However, in practice since D is not ideal, an additional regularizer is needed. In the paper we use relativistic loss as the regularizer, resulting in:
> (objective 2)   min KL(D(x) || D(G(z))) - KL(A_0 || D(G(z)))
> We also replace it with an alternative one, so that the objective becomes:
> (objective 3)    min KL(A_1 || D(G(z))) - KL(A_0 || D(G(z)))
>
> The quantitative results of these objectives on Cifar10 are:
>                          FID
> Objective1     36.73
> Objective2     34.59
> Objective3     36.21
> DCGAN           38.56
> WGAN-GP      41.86
> LSGAN            42.01
> HingeGAN     42.40
>
> We have also included in Appendix A the training curves of using objective 2 and objective 3 on CelebA. On both datasets, we can see without the relativistic loss, RealnessGAN can still outperform baselines.
>
> ** RealnessGAN vs WGAN-GP **
> On CIFAR10 we train a better version of RealnessGAN by increasing the number of outcomes while keeping other hyperparameters the same (random seeds, etc) as before. The updated results are:
>                               FID          SWD
> RealnessGAN    34.59         22.80
> WGAN-GP          41.86         28.17
> where RealnessGAN outperforms WGAN-GP significantly, especially on SWD, which is reported to be more informative than FID.
>
> As for the standard deviation, the number looks like RealnessGAN is not very stable. However, it is due to the high scores in the early stage of training. As shown in Figure 4, when trained sufficiently, RealnessGAN could consistently outperform WGAN-GP and other baselines.
>
> You may also notice in Table 1, RealnessGAN performs better with large margins on CelebA, a dataset that is more challenging than CIFAR10. To further show that, we also compute the FID of RealnessGAN on FFHQ, where all baselines fail to obtain good generators. RealnessGAN receives a FID score of *17.18*. For reference, our re-implemented StyleGAN when trained in the same setting receives a FID score of *16.12*. StyleGAN is one of the SOTA architectures. RealnessGAN using DCGAN leads to comparable results.
>
> All of the above results provide strong evidence on the effectiveness of RealnessGAN.

---

> > ### Comment · AnonReviewer1 · 2019-11-14
> > **Reply to Authors**
> >
> > First I would like to say that I greatly appreciate the clarity of the answer and the efforts that has been put in the rebuttal. I still have some minor comments:
> >
> > 1) Concerning $A_0$ and $A_1$:
> >
> > Thank you for the clarification, it's much clearer now. I have seen that you precise in the paper that the ground truth "are chosen to ressemble 2 normal distributions". This seems like a reasonable choice, but can you precise why this choice in particular, have you tried other shapes ? Also  can you precise (maybe in appendix) how you discretize the normal distributions and what are the different skew used in the experiments ? It seems from the paper that the choice of $A_0$ and $A_1$ is very important for the performance of the approach, I thus believe that it should be clearly explained how to set them up in the paper, so that the results of the paper are reproducible.
> >
> > 2) Concerning the standard deviation (STD)
> >
> > I didn't notice the first time that the STD was computed across different iteration in training. I think it would be more meaningful to compute the STD across different seeds.

---

> > > ### Author Response · Authors · 2019-11-15
> > > **Reply to Reviewer #1**
> > >
> > > Thank you for your valuable feedback, and we are glad to know our rebuttal could clarify your concerns.
> > >
> > > As for your minor comments:
> > >
> > > 1.First of all, as mentioned previously, we will release the code that ensures the reproducibility of our results, because we believe this paper provides several interesting directions for future work, both theoretically and technically. Releasing our code could bring interested people in the game quickly.
> > >
> > > We have included an illustrative figure in Appendix D for our choice of A0 and A1. We do not write a lot about our choice in the paper for we do not want to leave an impression that only the precise re-choice of our chosen A0 and A1 could lead to the results reported in the paper. In fact, we didn’t spend much effort on choosing A0 and A1. In our experience, we care more about the KL divergence between A0 and A1, for we found both theoretically and empirically it is important. And we only adjust their shapes when we feel it’s hard to find a specific pair that has sufficient KL divergence using current shapes.
> > >
> > > 2.Our way to compute STD follows previous works, such as [1]. And thank you for your suggestion, where we agree computing STD over random seeds is a good choice. Unfortunately, we cannot provide such statistics currently due to the time limit. We will certainly consider this suggestion later.
> > >
> > > [1] Jolicoeur-Martineau. The relativistic discriminator: a key element missing from standard GAN. ICLR 2019.

---

### Official Review · AnonReviewer2 · 2019-10-26
**Official Blind Review #2**

**Rating:** 8

**Review:**

The paper proposes to improve upon GANs by considering to infer the distribution of realness instead of binary true/false labels in the discriminator side. They carry out this idea with some theoretical arguments, and their method is shown to perform very well in empirical experiments on one synthetic dataset, and three real world datasets:CelebA, CIFAR-10, and FFHQ. Overall I feel this is a well presented paper with a simple yet interesting idea and solid results. The authors are encouraged to share their code and results to public.


**Experience Assessment:**

I have published one or two papers in this area.

**Review Assessment: Checking Correctness Of Derivations And Theory:**

I assessed the sensibility of the derivations and theory.

**Review Assessment: Checking Correctness Of Experiments:**

I assessed the sensibility of the experiments.

**Review Assessment: Thoroughness In Paper Reading:**

I read the paper at least twice and used my best judgement in assessing the paper.

---

> ### Author Response · Authors · 2019-11-12
> **Reply to Reviewer #2**
>
> Thank you for recognizing the value of our paper. And code will be released. We also include our revision in the paper (content in magenta).  Here we would like to update some new results.
>
> **FFHQ**
> Besides qualitative results included in the original version, we further compute FID as the quantitative result. Specifically, RealnessGAN yields a FID score of *17.18*. For reference, we also re-implement StyleGAN [1] and train it using the same setting, resulting in a FID score of *16.12*.
>
> While StyleGAN is regarded as an advanced structure that leads to SOTA results, RealnessGAN has obtained comparable results using the structure of DCGAN, suggesting that treating realness as a random variable is an effective approach.
>
> **The objective of G**
> In the original version, the relativistic loss is used as a regularizer.
> Specifically, we show in the theoretical analysis the ideal objective of G is:
> (objective 1)   min -KL(A_0 || D(G(z)))
> we show in the Appendix A such an overly-loosen objective could indeed lead to an acceptable generator. However, in practice since D is not ideal, an additional regularizer is needed. In the paper we use relativistic loss as the regularizer, resulting in:
> (objective 2)   min KL(D(x) || D(G(z))) - KL(A_0 || D(G(z)))
> We also replace it with an alternative one, so that the objective becomes:
> (objective 3)    min KL(A_1 || D(G(z))) - KL(A_0 || D(G(z)))
> The quantitative results of these objectives on Cifar10 are:
>                          FID
> Objective1     36.73
> Objective2     34.59
> Objective3     36.21
> DCGAN           38.56
> WGAN-GP      41.86
> LSGAN            42.01
> HingeGAN     42.40
> we have also included in Appendix A the training curves of using objective 2 and objective 3 on CelebA. On both datasets, we can see without the relativistic loss, RealnessGAN can still outperform baselines.

---

### Decision · Program_Chairs · 2019-12-19

**Decision:**

Accept (Spotlight)

**Comment:**

The paper proposes a novel GAN formulation where the discriminator outputs discrete distributions instead of a scalar. The objective uses two "anchor" distributions that correspond to real and fake data. There were some concerns about the choice of these distributions but authors have addressed it in their response. The empirical results are impressive and the method will be of interest to the wide generative models community.